# Temporal Dynamics of Bacterial Communities in *Ectropis grisescens* Following Cryogenic Mortality

**DOI:** 10.3390/insects16101040

**Published:** 2025-10-09

**Authors:** Xinxin Zhang, Zhibo Wang, Guozhong Feng, Qiang Xiao, Meijun Tang

**Affiliations:** 1Key Laboratory of Tea Quality and Safety Control, Tea Research Institute, Chinese Academy of Agricultural Sciences, Hangzhou 310008, China; zhangxinxin_19@163.com (X.Z.); xqtea@tricaas.com (Q.X.); 2State Key Laboratory of Rice Biology and Breeding, China National Rice Research Institute, Hangzhou 311401, China; fengguozhong@caas.cn

**Keywords:** *Ectropis grisescens*, bacteria communities, temporal dynamics, 16S rRNA sequencing

## Abstract

*Ectropis grisescens* is a major leaf-feeding pest in Chinese tea plantations, causing severe yield and quality losses. Current control methods, including chemical and biological agents, face challenges such as environmental risks and resistance. Bacteria play crucial roles in insect physiology and offer potential targets for sustainable pest management, yet their postmortem dynamics in *E. grisescens* remain poorly understood. This study used 16S rRNA sequencing to analyze postmortem bacterial community changes in *E. grisescens* cadavers at 0, 7, and 21 days after cryogenic mortality. We found that microbial diversity declined over time, while richness initially increased before decreasing. *Wolbachia*, the dominant endosymbiont, gradually disappeared after host death, whereas *Enterobacter* persisted as a major constituent. Non-dominant taxa such as *Lysinibacillus* and *Sporosarcina* temporarily increased by day 7 before returning to baseline levels. This research provides the first ecological profile of postmortem microbial succession in a lepidopteran pest, offering insights for developing novel pest control strategies.

## 1. Introduction

*Ectropis grisescens* (Warren; *Lepidoptera*: *Geometridae*) is among the most destructive leaf-feeding pests in Chinese tea plantations, capable of defoliating entire branches and severely impacting tea yield and quality [1,2]. *E. grisescens* damage tea plantations by feeding on leaves and young shoots, with severe infestations resulting in bare, scorched-looking branches [3]. Compared to other geometrid pests such as *Ectropis obliqua* and *Biston suppressaria*, *E. grisescens* has a wider geographical distribution and greater reproductive capacity, making it a major threat to tea production [4]. Current control strategies primarily depend on chemical pesticides, entomopathogenic viruses, and bacterial agents [5]. However, excessive use of chemical pesticides has resulted in environmental pollution and resistance development, whereas the effectiveness of biological agents, such as viruses and bacteria, remains inconsistent due to fluctuating environmental conditions and host immune responses [5]. Therefore, it is essential to develop sustainable and resilient pest management strategies that integrate multiple control approaches to ensure both agricultural productivity and ecological balance. Recent studies have increasingly highlighted the pivotal roles of endosymbiotic bacteria in insect physiology and adaptability, emphasizing their potential as novel targets for biological control. These microbes support host survival by supplying essential nutrients, synthesizing vitamins, and detoxifying harmful compounds such as pesticides and plant secondary metabolites [6]. Thereby enhancing nutrient absorption under conditions of dietary nutrient insufficiency; furthermore, they can provide protection against pathogens and support detoxification of pesticides or harmful plant secondary metabolites [7]. Shao et al. (2017) revealed a symbiotic mechanism in which *Enterobacter mundtii*, residing extracellularly in *Spodoptera littoralis*, selectively antagonizes intestinal pathogens by secreting narrow-spectrum antimicrobial peptides, thereby maintaining microbiome homeostasis and reducing the risk of enteric infection [8]. Insect endosymbionts can also enhance host adaptability, for example, by assisting lepidopteran larvae in breaking down plant cell walls and tolerate toxins. The research found that mulberry-derived 1-deoxynojirimycin (DNJ) inhibited six lepidopteran species but had no inhibitory effect on *Bombyx mori*. High-throughput sequencing identified *Pseudomonas fulva* ZJU1 as a dominant gut bacterium in DNJ-fed silkworms. This strain efficiently degraded DNJ in vitro, and when reintroduced into DNJ-exposed *Spodoptera exigua*, it restored larval growth, confirming its role in conferring toxin tolerance [9]. Additionally, *Wolbachia* strains can manipulate host reproductive strategies, such as inducing cytoplasmic incompatibility, to facilitate their own transmission. For instance, studies have shown that *Wolbachia* induces cytoplasmic incompatibility in host populations, thereby reducing their reproductive success [10]. These multifaceted interactions underscore the evolutionary significance of bacteria in insect ecology and highlight their potential as targets for novel pest management strategies.

Microbial community succession refers to the process by which microbial assemblages change over time. Following insect death, the microbial communities inhabiting the internal and external surfaces of the carcass undergo a series of orderly changes. This successional process is not only critical for nutrient cycling and energy flow within ecosystems, but also holds significant application potential in fields such as forensic science, agricultural waste management, and biological control [11,12,13]. In forensic science, microbial community succession provides novel approaches and insights for estimating the post-mortem interval (PMI). Studies have demonstrated that changes in the composition and structure of microbial communities are closely associated with the stage and duration of decomposition, enabling their use as a “microbial clock” for PMI estimation [14]. For instance, Metcalf et al., in a study using mouse models, found that microbial community data could accurately estimate PMI with an error margin within 2–3 days [14]. In the field of biological control, research on microbial community succession has informed new strategies for pest management. Studies indicate that manipulating the microbial communities of pests can influence their growth, development, and reproduction, thereby achieving control objectives [11]. For example, researchers have successfully suppressed the growth and reproduction of housefly larvae by inoculating them with specific microbial consortia, thereby reducing the reliance on chemical pesticides [11].

The gut microbiota of lepidopteran insects exhibits high diversity and is influenced by host diet, age, environment, and physiological state. Proteobacteria are commonly found in the gut of Lepidoptera, while Firmicutes—such as *Clostridium* and *Enterococcus*—are often dominant during the larval stage [15,16]. Other frequently observed taxa include *Bacteroidetes*, *Pseudomonas*, and *Bacillus*. Microbial transmission generally occurs through two main routes: vertical and horizontal transmission, with the dominant mode often depending on the host species and environmental conditions [17]. According to existing literature, the adult *E. grisescens* microbiota is predominantly composed of *Wolbachia* (28.97%), *Enterobacter* (24.17%), *Pseudomonas* (14.82%), *Arthrobacter* (7.74%), and *Melissococcus* (5.09%) [18]. To date, the transmission mechanisms of these microbial symbionts in *E. grisescens* have not been systematically investigated. Previous studies have identified two closely related species in the tea geometrid complex, namely *Ectropis obliqua* and *Ectropis grisescens* [4]. Compared to *Ectropis obliqua*, *Ectropis grisescens* exhibits a broader geographical distribution and demonstrates stronger resistance to the EcobNPV [19,20]. Previous studies have shown that *E. grisescens* harbors *Wolbachia*, whereas *E. obliqua* lacks this endosymbiont. Furthermore, *Wolbachia*-mediated cytoplasmic incompatibility (CI) has been shown to contribute to reproductive isolation between these two species [21,22]. Although some insights have been gained regarding the interaction between *Wolbachia* and two closely related species, the lack of fundamental studies hinders a deeper understanding of the functional roles of *Wolbachia* in these tea pests. For instance, it remains unclear whether horizontal transmission of *Wolbachia* occurs within *E. grisescens* populations, or whether there are interactions between *Wolbachia* and viruses infecting *E. grisescens* hosts. Therefore, we conducted this study to investigate the postmortem dynamics of the microbial community in *E. grisescens*. In this study, we explored high-throughput 16S rRNA gene sequencing to investigate temporal and compositional changes in the bacterial communities of *E. grisescens* following host mortality. Our results revealed significant shifts in the abundance and diversity of key bacterial taxa—such as *Wolbachia*, *Enterobacter*, and *Pseudomonas*—which are closely linked to host physiological processes [21]. These findings advance our understanding of host–symbiont dynamics, particularly the survival and succession of symbionts after host death, and suggest novel pest management strategies based on disrupting symbiotic relationships to impair *E. grisescens* fitness.

## 2. Materials and Methods

### 2.1. Insect Collection

*E. grisescens* were collected from Yueqing, Zhejiang Province, China (120.99° E, 29.50° N). The collected larvae were reared in the laboratory (25 ± 1 °C; 13 h light: 11 h dark photoperiod; relative humidity of 75–80%). Fresh leaves of the Yingshang tea cultivar were provided daily as the sole food source until the larvae were sampled for subsequent experimental.

### 2.2. Samples Preparation

Twelve *E. grisescens* larvae hatched on the same day were selected and reared indoors until they reached the third instar larval stage. At this point, the larvae were immediately immersed in liquid nitrogen (about 30 s) to ensure complete freezing of the core tissue, thereby rapidly halting metabolic activity. The samples were then divided into three groups, each comprising four larvae individually stored in 2 mL sterilized EP tubes. Accordingly, Group A and B were statically incubated in tubes placed in a laboratory incubator at 23 °C for 7 and 21 days, respectively, to simulate natural decomposition. After incubation, the samples were transferred to a −80 °C freezer for storage prior to subsequent analysis. The control group (CK) was directly stored at −80 °C without prior incubation.

The sampling time points (7 and 21 days post-infection) were selected based on the established pathology of EcobNPV infection in third-instar larvae. Seven days post-infection corresponds to the average latency period, coinciding with the initial onset of symptoms. Twenty-one days post-infection corresponds to the terminal stage, when larval disintegration is nearly complete and the carcass is laden with polyhedra. These time points were chosen to investigate bacterial community succession after death and to establish a baseline for future studies on bacterial dynamics during EcobNPV infection.

### 2.3. DNA Extraction

A sterile 4 mm steel bead was added to each 2 mL EP tube containing a sample, followed by homogenization using a tissue lyser for 2 min. The resulting tissue homogenate was used for DNA extraction following the manufacturer’s protocol of the DNeasy Blood and Tissue Kit (Qiagen, Hilden, Germany). Extracted DNA concentrations were measured using a NanoDrop 2000 (Thermo Fisher Scientific, Waltham, MA, USA), and the DNA samples were stored at −20 °C for further analysis.

### 2.4. PCR Amplification and Sequencing of Microbial 16S rDNA Genes

The V3–V4 hypervariable region of the 16S rRNA gene was amplified from the DNA samples using primers 341F (CCTACGGGNGGCWGCAG) and 805R (GACTACHVGGGTATCTAATCC) [23]. PCR was conducted using Phusion^®^ High-Fidelity PCR Master Mix (Thermo Fisher Scientific, Waltham, MA, USA) under the following conditions: initial denaturation at 94 °C for 3 min; 5 cycles of denaturation at 94 °C for 30 s, annealing at 46 °C for 20 s, and extension at 65 °C for 30 s; followed by 20 cycles of denaturation at 94 °C for 20 s, annealing at 55 °C for 20 s, and extension at 72 °C for 30 s; with a final extension at 72 °C for 5 min. PCR products (550 bp) were verified via electrophoresis on a 1% agarose gel and subsequently sequenced using the Illumina MiSeq platform (Illumina, San Diego, CA, USA) by Zhejiang Tianke Biotechnology Co., Ltd., Hangzhou, China.

### 2.5. Data Analysis

Raw reads were assembled using PEAR (v1.2.11) [24], followed by quality filtering to retain high-quality effective tags (quality score > 20, read length > 200 bp). High-quality tags were clustered into operational taxonomic units (OTUs) at 97% similarity using UPARSE (V8.1.1861) [25]. Representative OTUs were taxonomically classified by aligning them to the SILVA database (http://www.arb-silva.de, accessed 24 August 2025) using the UCLUST algorithm. Alpha diversity indices—including Observed-species, Chao1, Shannon, Simpson, ACE, and Good’s coverage, were calculated using QIIME software (Version 1.9.1) [26]. Statistical analyses were performed using SPSS v17.0 (IBM Corp., Armonk, NY, USA). One-way ANOVA followed by Tukey’s post hoc test was applied to compare α-diversity indices, with significance set at *p* < 0.05.

## 3. Results

### 3.1. Differential Assembly and Taxonomic Annotation of Symbiotic Bacterial Communities in Ectropis grisescens

High-throughput 16S rRNA gene was conducted on 12 samples across three treatment groups: a control group (CK) a 7-day static incubation group (Group A), and a 21-day static incubation group (Group B), with four replicates per group. The table of correspondence between BioSample IDs and samples of *E. grisescens* is shown in Appendix A. Sequencing results and quality control metrics are summarized in Table 1. All samples exhibited Q20 score above 98% and Good_coverage value exceeding 0.99, indicating high sequencing accuracy and comprehensive taxonomic coverage sufficient for downstream analyses.

A total of 582,490 raw reads were obtained, with an average length of 435 bp. After stringent quality filtering, 508,816 high-quality tags were retained for downstream analyses. Sequences were clustered into operational taxonomic units (OTUs) at a 97% similarity threshold, yielding 299 distinct OTUs. Taxonomic annotation revealed 12 phyla, 19 classes, 42 orders, 84 families, 151 genera, and 97 species across all samples. The CK group exhibited the highest taxonomic diversity, with annotations including 12 phyla, 19 classes, 42 orders, 79 families, 135 genera, and 90 species. In contrast, Group B showed the lowest diversity, with only 6 phyla, 10 classes, 20 orders, 30 families, 57 genera, and 35 species annotated.

### 3.2. Differences in Diversity and Richness of Bacterial Communities in Ectropis grisescens Under Different Treatments

The diversity and richness of bacterial communities were evaluated using alpha diversity indices (Figure 1). Shannon and Simpson indices were used to assess microbial diversity, with higher values indicating greater community diversity. Similarly, ACE and Chao1 indices were used to estimate microbial richness, with higher values indicating greater richness.

According to the Shannon index analysis, the diversity in Group A (1.24) after 7 days of incubation and Group B (0.67) after 21 days was lower than that of the control group (CK, 1.36). Notably, Group B showed a statistically significant decrease in diversity compared to the control group. The Simpson index indicated that Group A (0.44) was slightly higher diversity than the control group (CK, 0.36), though the difference was not statistically significant. However, Group A exhibited significantly higher diversity than Group B (0.19). Ace and Chao1 indices showed consistent trends across the three groups, with no statistically significant differences observed. Group A exhibited the highest richness values (167.65 for ACE and 148.24 for Chao1, respectively), while Group B exhibited the lowest values (133.58 and 126.29, respectively).

A comprehensive analysis indicated that the diversity of bacterial communities in *E. grisescens* decreased over time following cold-induced mortality, whereas microbial richness initially increased before declining at later stages.

### 3.3. Similarity Analysis of Bacterial Communities in Ectropis grisescens Under Different Treatments

Principal Coordinates Analysis (PCoA) based on OTU annotations was conducted to evaluate the similarity of bacterial communities among the 12 samples (Figure 2). The PERMANOVA test based on the Bray-Curtis distance measures showed that the bacterial community structure was significantly (*p* < 0.01) different among these clusters. In the PCoA plot, each point represents a sample, and the distance between points reflects the degree of similarity in their symbiotic bacterial compositions. The analysis revealed that, except for sample CK1 from the control group (CK) and sample A3 from treatment group A, samples within each group clustered into distinct regions, indicating high intra-group consistency in symbiotic bacterial composition.

Specifically, most samples from the control group (CK) were located in the fourth quadrant, whereas samples from treatment groups A and B predominantly clustered in the first quadrant. This spatial distribution indicates that the bacterial community structures of groups A and B were highly similar to each other but distinctly distinct from that of the control group (CK). These findings underscore significant differences in symbiotic bacterial compositions between the treatment groups and the control, as well as the internal consistency of microbial communities within each treatment group.

### 3.4. Analysis of Composition and Abundance of Microbial Communitys in Ectropis grisescens Under Different Treatments

High-throughput sequencing revealed significant differences in microbial community composition among the different treatment groups of *E. grisescens*. The number of operational taxonomic units (OTUs) identified in each group was as follows: CK (250 OTUs), Group A (171 OTUs), and Group B (137 OTUs). Among these, 88 core OTUs were consistently shared across all three treatment groups (Figure 3). Comparative analysis showed that group A shared 129 OTUs with the CK group, contained 43 unique OTUs, and lost 121 OTUs. Similarly, group B shared 97 OTUs with the CK group, contained 40 unique OTUs, and lost 153 OTUs. These results suggest a time-dependent reduction in the diversity of microbial communities following the death of *E. grisescens*, along with the colonization of new microbial taxa within the host cadaver.

At the phylum level (Figure 4), 12 phyla were annotated in the CK group, ranked in descending order of relative abundance: Proteobacteria, Firmicutes, Bacteroidetes, Actinobacteria, Fusobacteria, Epsilonbacteraeota, Patescibacteria, Gemmatimonadetes, Spirochaetes, Deinococcus-Thermus, Cyanobacteria, and Tenericutes. In contrast, Group A and Group B exhibited annotations for 8 and 6 of these phyla, respectively. Across all three groups, the microbial community was predominantly composed of *Proteobacteria*, *Firmicutes*, and *Bacteroidetes*, which together accounted for over 99% of total abundance, indicating a consistent proportional pattern.

Specifically, *Proteobacteria* was the most abundant phylum, with relative abundance of 95.95% in the CK group, 91.96% in Group B, and 73.75% in Group A. Conversely, Firmicutes showed the highest relative abundance in Group A (26.00%), followed by Group B (7.80%), and was least abundant in the CK group (2.38%). Notably, several phyla, including *Gemmatimonadetes*, *Deinococcus-Thermus*, and *Cyanobacteria*, were absent in either Group A or Group B.

At the genus level (Figure 5), the composition of microbial communities varied significantly among the treatment groups. In the CK group, the top three genera by relative abundance were *Wolbachia* (78.04%), *Enterobacter* (12.80%), and *Acinetobacter* (1.28%). In Group A, the predominant genera were *Enterobacter* (70.61%), *Lysinibacillus* (18.59%), and *Sporosarcina* (6.81%). Similarly, in Group B, the dominant genera were *Enterobacter* (89.79%), *Lysinibacillus* (4.94%), and *Sporosarcina* (2.52%). These findings highlight notable shifts in bacterial community structure across treatment groups, with Enterobacter emerging as the dominant genus in Group A and Group B, and *Wolbachia* predominanting in the control group.

### 3.5. Temporal Dynamics of Bacteria in Ectropis grisescens Following Liquid Nitrogen-Induced Mortality

The composition of endosymbiotic bacterial communities in *E. grisescens* exhibited marked temporal changes following host mortality. *Wolbachia*, the dominant bacterial genus in living *E. grisescens*, declined progressively after host death, suggesting that its survival may depend on nutrients supplied by the living host. In contrast, *Enterobacter*, another predominant genus in *E. grisescens*, proliferated over time within the host carcass and eventually became the dominant bacterial taxon.

Notably, *Lysinibacillus* and *Sporosarcina*, which were not dominant in living *E. grisescens*, exhibited increased relative abundance seven days post-mortem. However, by day 21, their relative abundances declined to levels comparable to those observed in the control group (Figure 6). These findings highlight the dynamic succession of bacterial communities following host death and suggest distinct ecological roles and survival strategies among different bacterial taxa.

## 4. Discussion

*E. grisescens* is a major pest in tea plantations in China, and its infestation can cause significant economic losses to tea production. *Endosymbiotic* bacteria in insects play critical roles in supplying essential nutrients, detoxifying harmful substances, and enhancing host survival, thereby offering considerable potential for biological pest control [6]. In this study, we investigated the temporal dynamics of bacterial communities in *E. grisescens* by sequencing the microbiota of host carcasses at 7 and 21 days postmortem. Our results revealed that the cryogenic mortality of *E. grisescens* induced significant temporal shifts in its community structure. Microbial diversity declined over time, whereas richness initially increased and then decreased (Figure 1). Notably, the bacterial community compositions in postmortem samples (Groups A and B) diverged markedly from those in the control (CK), with dynamic shifts in dominant taxa such as *Wolbachia*, *Enterobacter*, *Lysinibacillus*, and *Sporosarcina (*Figure 5).

*Wolbachia* has long been a focal point of research in the studies on the two closely related geometrid moth species, *E. grisescens* and *E.obliqua. Wolbachia*, an obligate intracellular endosymbiont widely distributed in arthropods and nematodes, depends on host-derived nutrients and cellular machinery for replication and long-term persistence [27]. *Wolbachia* orchestrates a range of reproductive manipulations in insect hosts, including parthenogenesis induction, feminization, male killing [28,29], and cytoplasmic incompatibility [22]. Moreover, *Wolbachia* exhibits dual functional roles in host protection and nutritional regulation within symbiotic systems [30]. In *Drosophila*, *Wolbachia* specifically inhibits the RNA replication phase of *Semliki Forest virus* (SFV), thereby suppressing viral protein expression [31]. In *Aedes aegypti* mosquitoes infected with *Wolbachia*, the production of antimicrobial peptides (e.g., defensin, cecropin) is upregulated via aae-miR-34-3p (a *Aedes aegypti*-derived microRNA) -mediated activation of the Toll pathway, resulting in significantly reduced *dengue virus* (DENV) titers and enhanced viral resistance [32]. Experimental evidence demonstrates that *Wolbachia* provides its nematode with essential metabolites, including riboflavin, heme, glutathione, and key precursors for purine and pyrimidine biosynthesis [33]. Depletion of *Wolbachia* severely impairs host fitness, manifested as growth retardation, abnormal embryogenesis, and increased mortality [34]. Notably, *Wolbachia* actively modulates host iron homeostasis. Under iron-deficient conditions in *Drosophila melanogaster*, *Wolbachia* infection enhances female fecundity by promoting iron utilization [35]. More strikingly, in *Asobara tabida* wasps, *Wolbachia* maintains iron homeostasis by suppressing ferritin overexpression, an imbalance that would otherwise trigger apoptotic cell death in uninfected individuals, thereby ensuring germline survival and reproductive success [36]. In this study, *Wolbachia* was identified as the dominant endosymbiont in *E. grisescens*; however, its abundance declined significantly following host mortality (Figure 6). This phenomenon has not been formally documented in other species to date. This observation suggests a nutritional dependency of *Wolbachia* on its lepidopteran host and points to potential host–symbiont metabolic cross-talk warranting further mechanistic investigation. As the corpse decomposes, *Wolbachia* gradually disappears, indicating that it is difficult for *Wolbachia* to exist independently outside its host in nature. Therefore, the disappearance of *Wolbachia* suggests that there may be no horizontal transmission mechanism of this bacterium in the species *E. grisescens*.

In contrast, *Enterobacter* persisted as a dominant taxon across all experimental groups, likely due to its metabolic versatility and tolerance to alkaline environments (Figure 6) [37,38]. *Enterobacter* can support host survival by provisioning essential nutrients such as amino acids and B-vitamins and by contributing to key metabolic processes [39,40]. It can also protect hosts by inhibiting pathogen colonization and by enhancing or priming host immune responses [41,42]. Neither *Lysinibacillus* nor *Sporosarcina* were dominant bacterial taxa in *E. grisescens* under normal conditions. However, their relative abundances exhibited a transient increase at 7 days postmortem, followed by a decline to baseline levels by day 21 (Figure 6). This temporal pattern suggests opportunistic proliferation of these taxa during a specific decomposition process. Species of *Lysinibacillus* and *Sporosarcina* possess sporulation capabilities and notable resistance to environmental stress, including moderate cold tolerance. Under cold conditions, these bacteria adapt via mechanisms such as modulating membrane fluidity and synthesizing cryoprotective proteins, allowing them to remain dormant until environmental conditions become favorable for survival [43]. During insect carcass decomposition, *Lysinibacillus* and *Sporosarcina* utilize nutrients derived from the host body to support growth and reproduction. These bacteria proliferate rapidly during the early stages of decomposition when nutrient availability is high, contributing to material cycling by breaking down complex organic compounds into simpler inorganic forms that are subsequently recycled into the ecosystem [44]. These insights hold promising implications for the development of novel pest management strategies. For instance, the transient colonization of *Lysinibacillus* and *Sporosarcina* species after host death contributes to cadaver decomposition and interspecific resource competition. This ecological observation suggests a potential pest management approach in which the introduction of these competitive probiotic strains following application of *Ectropis obliqua* nucleopolyhedrovirus (EcobNPV) could accelerate the decomposition of deceased *E. grisescens* individuals. Such an approach could effectively enhance pathogen transmission within the pest population by increasing cadaver turnover. The observed decline in their abundance by 21 may be attributed to either resource depletion or competitive inhibition by other microbial populations, such as *Enterococcus* spp. Our findings align with previous studies on insect-microbe systems, which report that host mortality induces significant alterations in microbial community structure [45,46].

In this study, we investigated the temporal dynamics of bacterial communities in *E. grisescens* following mortality. Our findings revealed a progressive decline in microbial diversity over time, whereas bacterial richness initially increased and then decreased. From an ecological perspective, the collapse of *Wolbachia* underscores the vulnerability of obligate symbionts under host-derived stress, suggesting a potential nutritional dependency of *Wolbachia* on its lepidopteran host. In contrast, facultative symbionts such as *Enterobacter* exhibit ecological plasticity, enabling their persistence during both host life and postmortem stages. The recruitment of environmental bacteria (e.g., *Lysinibacillus* spp.) during decomposition suggests that insect cadavers may serve as transient microbial reservoirs, potentially influencing nutrient cycling and horizontal gene transfer within ecosystems. This study has several limitations. First, although amplicon sequencing provides valuable insights into successional patterns of microbial communities during decomposition, inferences regarding functional roles remain speculative in the absence of corroborating evidence from meta-transcriptomic or culturomic approaches. Future work will focus on isolating key bacterial taxa identified herein and conducting precise functional validation in laboratory simulation systems. Secondly, the decomposition of *E. grisescens* in this study was conducted under controlled laboratory conditions (constant 23 °C). Although not a completely sterile environment, this setting still differs significantly from natural habitats, where abiotic factors and stochastic microbial introductions play critical roles. Therefore, future research should prioritize investigating decomposition dynamics under natural environmental conditions to enhance the ecological relevance and translational validity of the findings. This study highlights the dynamic and complex interactions between *E. grisescens* and its bacteria following host death. These findings contribute to a broader understanding of host-microbe interactions and microbial succession in decomposing systems, offering potential insights for future pest control strategies and microbial ecology research.

## 5. Conclusions

This study provides a comprehensive characterization of the postmortem microbial succession in *Ectropis grisescens* cadavers following cryogenic mortality, revealing dynamic temporal changes in bacterial community structure. We observed a general decline in microbial diversity over time, while species richness exhibited an initial increase prior to subsequent depletion. Notably, the dominant endosymbiont *Wolbachia* gradually disappeared after host death, whereas *Enterobacter* persisted as a major constituent. Non-dominant taxa such as *Lysinibacillus* and *Sporosarcina* showed transient increases in abundance at day 7 before returning to baseline levels by day 21.

These findings offer the first ecological atlas of postmortem microbial succession in a lepidopteran pest, enhancing our understanding of host–microbe interactions beyond host death. The temporal dynamics of key symbionts, especially the decline of *Wolbachia* and the resilience of *Enterobacter*, may have implications for the efficacy of biocontrol strategies and the manipulation of insect microbiomes. Future studies should focus on elucidating the functional roles of these shifting microbial communities and exploring their potential in the development of novel, sustainable pest management practices.

## Figures and Tables

**Figure 1 insects-16-01040-f001:**
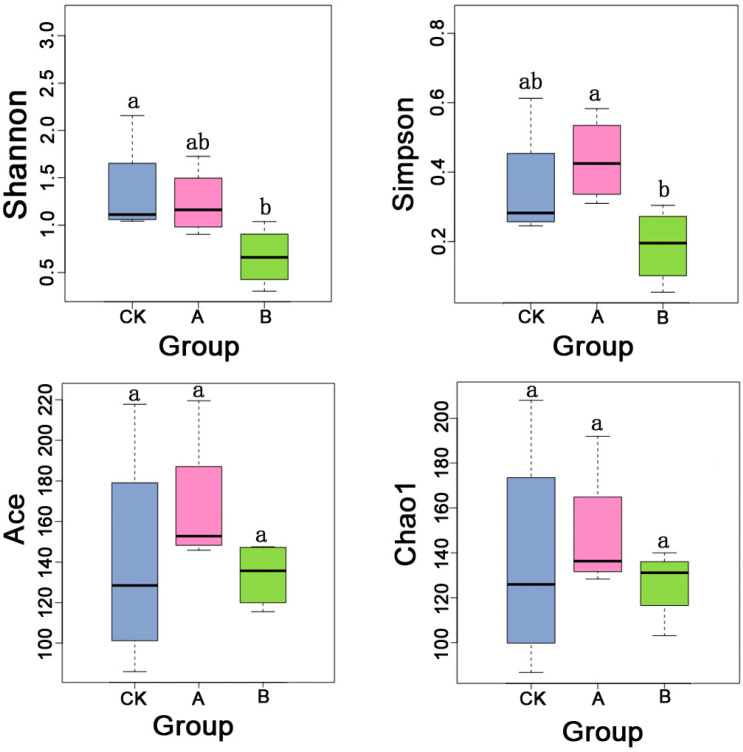
Diversity and richness of symbiotic bacteria in *Ectropis grisescens* under different post-mortem treatments. Alpha diversity was evaluated using the Chao1, Shannon, Simpson, and ACE indices. Data were analyzed using SPSS Statistics 17.0 (IBM). Error bars represent ± SEM. CK (control, directly stored at −80 °C); A (7 days incubation at 23 °C, then stored at −80 °C); B (21 days incubation at 23 °C, then stored at −80 °C). Different lowercase letters indicate statistically significant differences (*p* < 0.05, ANOVA with post hoc test); groups sharing the same letter are not significantly different, and “ab” denotes no significant difference from either “a” or “b”.

**Figure 2 insects-16-01040-f002:**
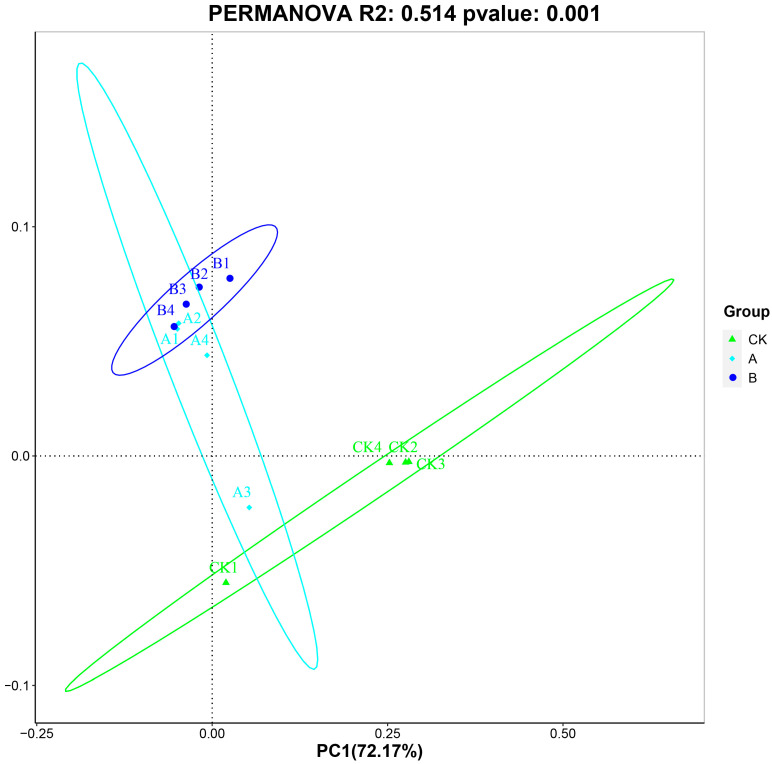
Principal Coordinate Analysis plot of microbial communities across 12 samples based on Operational Taxonomic Units (OTUs). Principal Coordinate Analysis (PCoA) plot demonstrating the variation among samples. The percentage of total variance explained by each principal coordinate is indicated on the corresponding axis (PC1: 72.17%; PC2: 12.42%). Axes represent standardized relative distances. Proximity between points indicates similarity in community composition.

**Figure 3 insects-16-01040-f003:**
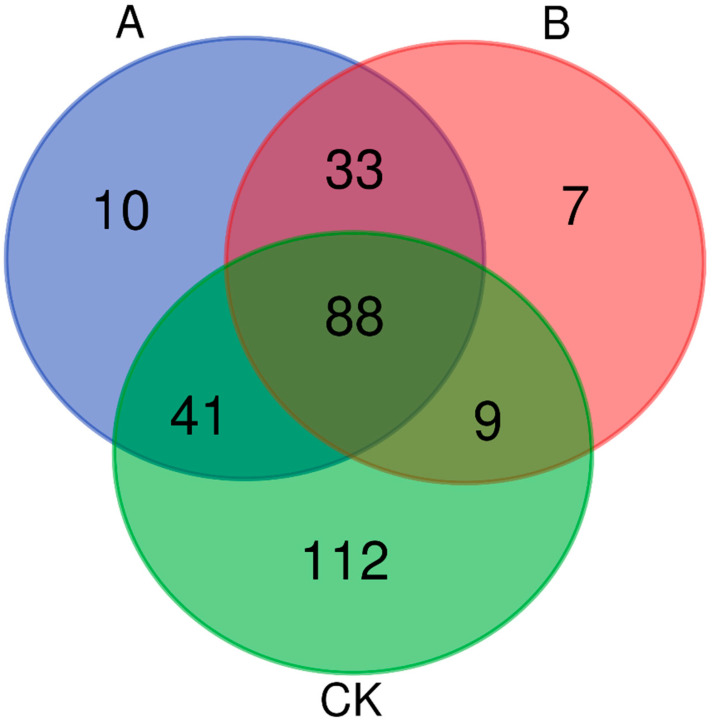
Venn Diagram Showing the Distribution of Shared and Unique OTUs among Different Treatments of *Ectropis grisescens* Based on High-Throughput Sequencing. The numbers within each section of the diagram represent the counts of OTUs that are either unique to a specific treatment group or shared among multiple treatment groups. Each circular area represents a distinct treatment group, with overlapping regions indicating the number of shared OTUs among groups and non-overlapping regions indicating the number of unique OTUs within each group. The treatment groups include Group A, Group B, and the control group (CK). The data were generated through high-throughput sequencing and classified into OTUs at a 97% similarity threshold using the Uparse software (v8.1.1861).

**Figure 4 insects-16-01040-f004:**
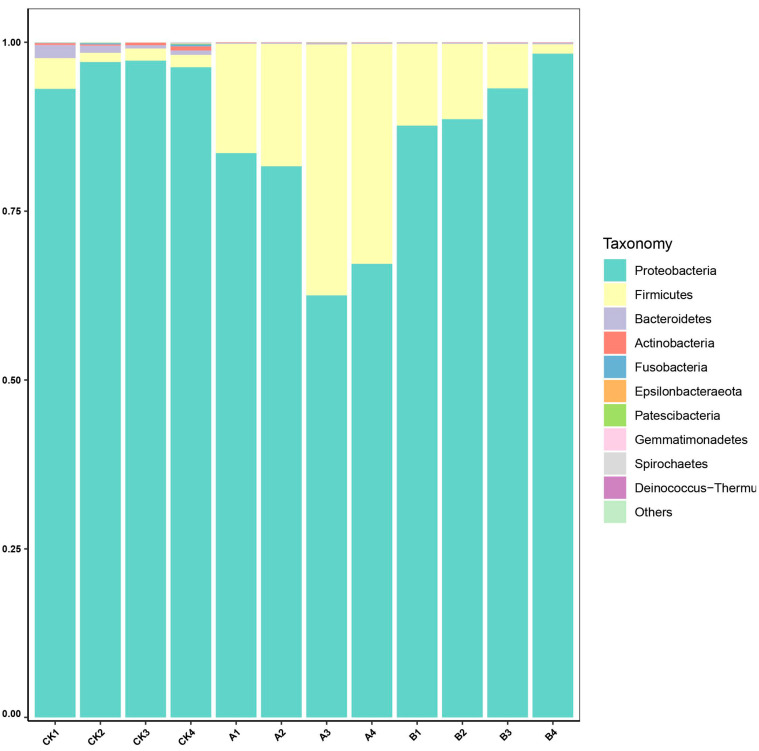
Relative abundance of microbial phyla in *Ectropis grisescens*. The stacked bar chart illustrates the composition of microbial communities at the phylum level, analyzed through high-throughput sequencing of 16S rRNA gene amplicons (V3–V4 region). Bars of different colors represent the abundance of the top ten microbial phyla. “Others” indicates the combined abundance of microbial phyla not included in the top ten.

**Figure 5 insects-16-01040-f005:**
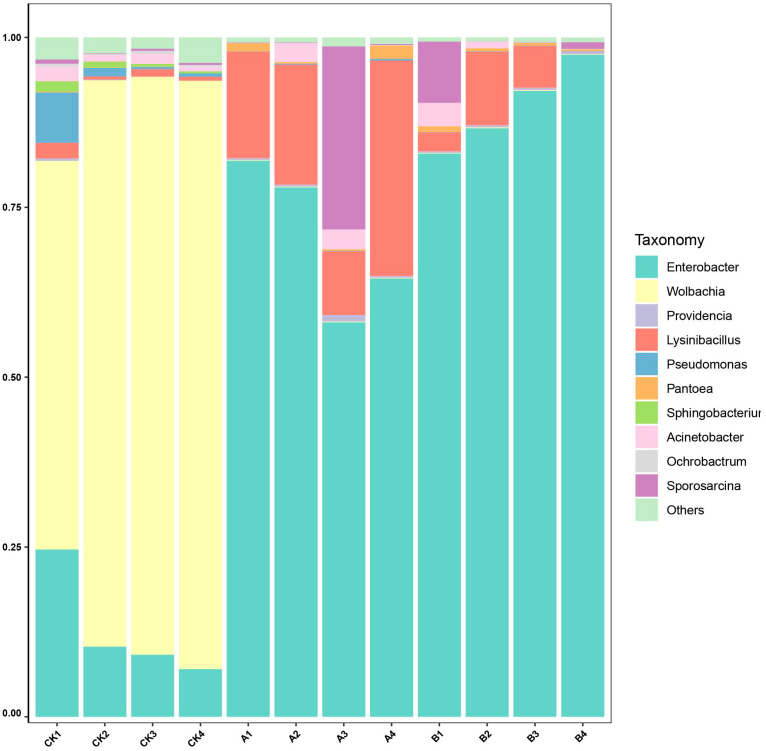
Relative abundance of dominant microbial genera in *Ectropis grisescens*. The stacked bar chart illustrates the composition of microbial communities at the phylum level, analyzed through high-throughput sequencing of 16S rRNA gene amplicons (V3–V4 region). Bars of different colors represent the abundance of the top ten microbial genera. “Others” indicates the combined abundance of microbial genera not included in the top ten. Columns of different colors represent dominant genera (relative abundance > 1%).

**Figure 6 insects-16-01040-f006:**
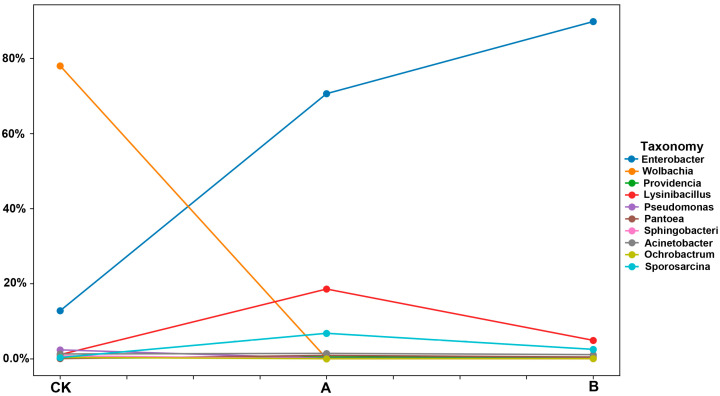
Temporal changes in the internal microbial communities of *Ectropis grisescens* following mortality. The broken lines of different colors represent the abundance of the top ten microbial genera. “Others” indicates the combined abundance of microbial genera not included in the top ten. The vertical axis represents the expression abundance, which is shown as a percentage.

**Table 1 insects-16-01040-t001:** Quality Control and Data Statistics for Sequence Processing. Raw_PE: Total raw paired-end reads generated by sequencer, including low-quality sequences and adapters. Effective tags: High-quality sequences after filtering out low-quality bases (Q < 20), short reads, and chimeras. OTUs: Operational Taxonomic Units clustered at 97% sequence similarity, representing microbial species. Q20/%: Percentage of bases with ≤1% error rate (Phred score ≥ 20).

Group	Raw_PE	Effective Tags	OTUs	Q20/%	Goods_Coverage
CK	209,019	193,753	250	99.05	0.999
A	181,515	152,690	171	98.87	0.999
B	191,956	162,373	137	98.85	0.999

## Data Availability

The original sequencing data presented in this study are openly available in the NCBI Sequence Read Archive (SRA) under accession number PRJNA1275583. [NCBI Sequence Read Archive (SRA) https://www.ncbi.nlm.nih.gov/sra?term=PRJNA1275583&cmd=DetailsSearch, accessed on 24 August 2025].

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
