# Peer review of "Temporal Dynamics of Bacterial Communities in Ectropis grisescens Following Cryogenic Mortality"

_insects, 2025, doi:10.3390/insects16101040_

Round 1

Reviewer 1 Report

Comments and Suggestions for Authors

The manuscript, “Temporal Dynamics of Endosymbiont Communities in Ectropis grisescens Following Cryogenic Mortality” compares the bacterial community in E. griescens at 0, 7, and 21 d after cryogenic mortality and identifies changes in microbial diversity over time. Overall, the data support the conclusions that microbial diversity increases at Day 7 then declines by Day 21. However, the significance of the results are not clear, especially considering the artificial conditions used for decomposition.

There are a number of major issues that complicate interpretation of the results and relevance of the findings to natural conditions:

Introduction

  1. It would be good to have background information on how the microbiome assembles in lab-raised moths. Are microbes acquired from diet or vertically transmitted?

Methods

  1. More detailed justification for why the Day 7 and Day 21 time points were chosen should be provided.
  2. Please provide detail on what “rapid frozen” (Line 127) entails – how much time? How does rapid freezing alter the microbiome profile? Comparing Day 0 frozen vs non-frozen would be informative to differentiate the effects of death/ decomposition on the microbiome from that of rapid freezing.

Results

  1. For Venn diagram analysis, was there a cut-off used (e.g., top 100 taxa or taxa found in all samples) when analyzing overlapping OTUs?
  2. Not informative to show both phyla and genera for the relative abundance plots. I suggest showing only genera.
  3. Fig 1, 2: text is very small, hard to read
  4. Fig 6: text is difficult to read; Chinese characters are in the figure legend.

Discussion

  1. Samples are decomposing in an enclosed, controlled environment (sterilized tube) with no exposure to abiotic changes and environmental microbes. It is difficult to extrapolate the current findings to the dynamics of carcasses in the natural environment. Limitations of the current findings should be discussed.
  2. Line 389: It is not clear from this manuscript how Lysinibacillus and Sporosarcina contribute to cadaver decomposition. A proper experiment would compare decomposition of cadavers with or without these bacterial taxa and also compare nutrient profile (e.g., proteins, lipids) of the cadaver.
  3. Amplicon sequencing does not differentiate between live microbes and inactive microbes. This point should be mentioned in the discussion. Decomposing carcasses would need to be plated on media to test whether microbes are viable.
  4. The rationale for this study could be more clearly stated. From the discussion, it seems that identifying microbes that contribute to Ectropis decomposition can help with population control methods that use Ecob-NPV. The connection between the application of Ecob-NPV and accelerated Ectropis decomposition is not clear – why is it advantageous for carcasses to decompose faster?
  5. 410 – considering that the carcasses decompose in a tube, it’s not clear that environmental bacteria are recruited.

There are a number of typos:

Throughout the manuscript text and figure legends, please italicize bacterial taxa.

20-21: Italicize bacterial taxa

58: “integrbreakingupltiple” – please correct

61: “nbreakingtabreaking” – please correct

65: synthesizing vitamins already mentioned in 62

67: detox already mentioned in line 62

168: the Day 0 group is not really a blank control group – “blank” would imply that it’s a background control with no DNA

301 (figure legend) – genera, not phyla

346, 351: italicize Wolbachia and Aedes aegypti

352: What is “aae-miR-34-3p”?

355: “imbal”? “imbalancet”?

Author Response

Comments 1: Introduction  It would be good to have background information on how the microbiome assembles in lab-raised moths. Are microbes acquired from diet or vertically transmitted?

Response 1: We thank the reviewer for this valuable suggestion. The gut microbiota of lepidopteran insects exhibits high diversity and is influenced by host diet, age, environment, and physiological state. Proteobacteria are commonly found in the gut of Lepidoptera, while Firmicutes—such as Clostridium and Enterococcus—are often dominant during the larval stage. Other frequently observed taxa include Bacteroidetes, Pseudomonas, and Bacillus. Microbial transmission generally occurs through two main routes: vertical and horizontal transmission, with the dominant mode often depending on the host species and environmental conditions. 

To date, the modes of microbial transmission in Ectropis grisescens have not been systematically investigated. In our previous study, through controlled single-factor experiments and genetic crosses, we confirmed that Wolbachia is vertically transmitted in this species. Although horizontal transmission of Wolbachia has been reported in other insects, such as Bemisia tabaci and Tetranychus turkestani, whether it occurs in E. grisescens remains unknown. Our results provide evidence that the horizontal transmission pathway of Wolbachia does not exist in Ectropis grisescens

Comments 2: Methods:More detailed justification for why the Day 7 and Day 21 time points were chosen should be provided.

Response 2: Thank you very much for your suggestion. The selection of these two time points is based on the progression of EcobNPV infection in Ectropis grisescens. In this experiment, larvae were initially sampled at the 3rd instar. The 7-day interval corresponds to the average latent period of EcobNPV in 3rd-instar larvae, marking the onset of infection symptoms. The 21-day point represents the stage at which infected 3rd instar larvae are nearly completely disintegrated, characterized by extensive polyhedra formation and mortality. These time points were chosen to investigate post-mortem bacterial changes in E. grisescens and to establish a foundation for future studies on bacterial dynamics during EcobNPV infection.

Comments 3: Please provide detail on what “rapid frozen” (Line 127) entails – how much time? How does rapid freezing alter the microbiome profile?

Response 3: Thank you for this valuable suggestion. In this study, "rapid freezing" refers to the process of immersing insects in liquid nitrogen for 30 seconds. This method instantaneously halts metabolic activity, thereby “freezing” the in vivo microbiome profile at the moment of death and preventing its degradation. Following this step, all samples were stored at –80°C, which is a standard protocol in microbiology for preserving microbial community integrity.

Comments 4: Comparing Day 0 frozen vs non-frozen would be informative to differentiate the effects of death/ decomposition on the microbiome from that of rapid freezing.

Response 4: We thank the reviewer for this valuable comment. In our experimental design, the Day 0 frozen sample (CK) is intended to represent the starting community state immediately upon death. As rapid freezing is our method of euthanasia, we focused our comparisons on the time-series data after death to elucidate the temporal dynamics. Therefore, we did not include an additional unfrozen control at Day 0.

Comments 5: Results For Venn diagram analysis, was there a cut-off used (e.g., top 100 taxa or taxa found in all samples) when analyzing overlapping OTUs? Not informative to show both phyla and genera for the relative abundance plots. I suggest showing only genera.

Fig 1, 2: text is very small, hard to read

Fig 6: text is difficult to read; Chinese characters are in the figure legend.

Response 5: We sincerely thank the reviewer for this valuable suggestion. In the Venn diagram analysis, we included all annotated OTUs. For a comprehensive view across different taxonomic resolutions, relative abundance plots are provided at both the phylum and genus levels, which aligns with standard practices in the field. Should space considerations or clarity require it, we are open to omitting the phylum-level plot.

We have adjusted the font sizes of the labels in Figures 1, 2, and 6 to ensure that all information is clearly legible.

Comments 6: Discussion Samples are decomposing in an enclosed, controlled environment (sterilized tube) with no exposure to abiotic changes and environmental microbes. It is difficult to extrapolate the current findings to the dynamics of carcasses in the natural environment. Limitations of the current findings should be discussed. 

Response 6: We appreciate the reviewer's suggestion. The use of sterile tubes was primarily to facilitate the freezing process in liquid nitrogen and to minimize potential confounding factors. It is important to note that the subsequent 7-day and 21-day incubation periods were not conducted under sealed or sterile conditions. In fact, our experimental intent was to simulate relevant environmental conditions to the greatest extent possible under specific, yet consistent, background conditions across all three treatment groups.

Comments 7: Line 389: It is not clear from this manuscript how Lysinibacillus and Sporosarcina contribute to cadaver decomposition. A proper experiment would compare decomposition of cadavers with or without these bacterial taxa and also compare nutrient profile (e.g., proteins, lipids) of the cadaver.

Response 7: Many thanks for your valuable suggestion. We fully agree that unequivocally confirming the specific roles of these two genera in decomposition would require comparing decomposition processes and nutritional profiles (e.g., proteins, lipids) of remains in the presence versus absence of these bacteria. In this preliminary study, our primary aim was to describe the successional patterns of the microbial community during decomposition using amplicon sequencing. Based on the strong correlation between their abundance dynamics and decomposition stages, we inferred their potential important roles. For instance, their dominance in the later stages of decomposition suggests that they may play key roles in breaking down recalcitrant proteins or lipids. We acknowledge that confirming their specific functions is a limitation of the current study, but it also represents a key focus of our future research. We have added this limitation, along with the excellent experimental approach proposed by the reviewer, as a direction for future investigations in the Discussion section. We plan to isolate these key bacterial strains from the samples and conduct precise functional validation in laboratory simulation systems, which will serve as a logical extension and deepening of this study.

Comments 8: Amplicon sequencing does not differentiate between live microbes and inactive microbes. This point should be mentioned in the discussion. Decomposing carcasses would need to be plated on media to test whether microbes are viable.

Response 8: Many thanks for your valuable suggestion. You are absolutely correct that DNA-based amplicon sequencing cannot distinguish between DNA signals derived from living cells and those from dead or inactive cells. Following your advice, we have now explicitly stated this technical limitation in the Discussion section. Furthermore, we have also mentioned that future studies could employ meta-transcriptomics or culturomics approaches to focus specifically on the metabolically active microbial communities during decomposition, thereby more accurately elucidating their functions. Your insightful comment has significantly enhanced the depth and rigor of our discussion.

Comments 9: The rationale for this study could be more clearly stated. From the discussion, it seems that identifying microbes that contribute to Ectropis grisescens decomposition can help with population control methods that use EcobNPV. The connection between the application of EcobNPV and accelerated Ectropis grisescens decomposition is not clear – why is it advantageous for carcasses to decompose faster? 

Response 9: We sincerely appreciate the reviewer for highlighting this important issue. As rightly pointed out, the proposed link between "microbial interventions" and "enhanced viral transmission or increased host mortality" mentioned in our initial manuscript lacks sufficient experimental support at this stage. This aspect will constitute a major direction of our subsequent research. Work in this area, particularly concerning viral mechanisms, has already been commenced, and the resulting findings will be reported in a future separate article.

Comments 10: 410 – considering that the carcasses decompose in a tube, it’s not clear that environmental bacteria are recruited.

Response 10: We thank the reviewer for this insightful comment. The use of sterile tubes was primarily intended to prevent contamination from external contact, thereby eliminating potential experimental interference, while also facilitating the operation of freezing-induced euthanasia. As for the post-euthanasia static placement, it did not actually occur in a sterile environment. This step was designed to simulate the natural decomposition process under laboratory conditions. However, it is important to note that laboratory conditions differ from those in the natural environment—for instance, they lack variable factors such as rainfall and intense sunlight. Instead, the process was conducted under constant temperature and humidity. This setup allows us to more accurately observe changes in the microbial community within Ectropis grisescens at different time points after death.

Comments 11: There are a number of typos: Throughout the manuscript text and figure legends, please italicize bacterial taxa.

20-21: Italicize bacterial taxa

58: “integrbreakingupltiple” – please correct

61: “nbreakingtabreaking” – please correct

65: synthesizing vitamins already mentioned in 62

67: detox already mentioned in line 62

301 (figure legend) – genera, not phyla

346, 351: italicize Wolbachia and Aedes aegypti

355: “imbal”? “imbalancet”?

Response 11: Thank you very much for your suggestions. I have revised the manuscript according to the formatting and textual errors you pointed out.

Comments 12: 168: the Day 0 group is not really a blank control group – “blank” would imply that it’s a background control with no DNA

Response 12: We thank the reviewer for this valuable suggestion. We have revised the manuscript by removing the term "blank" in reference to the Day 0 group, as appropriately suggested.

Comments 13: 352: What is “aae-miR-34-3p”?

Response 13: Among them, in Line 352, the phrase “‘aae-miR-34-3p’ is an endogenous microRNA (miRNA) in Aedes aegypti” has been modified to improve clarity:

In Aedes aegypti mosquitoes infected with Wolbachia, the production of antimicrobial peptides (e.g., defensin, cecropin) is upregulated via aae-miR-34-3p (a Aedes aegypti-derived microRNA) -mediated activation of the Toll pathway, resulting in significantly reduced dengue virus (DENV) titers and enhanced viral resistance.

Reviewer 2 Report

Comments and Suggestions for Authors

Zhang et al. investigated the bacterial communities of the leaf-feeding pest Ectropis grisescens after cryogenic mortality using 16S rRNA gene sequencing. The work provides valuable ecological insights into microbial succession in a major lepidopteran pest and highlights potential implications for pest management. Overall, the manuscript is well-structured, and the analytical approaches are generally appropriate. The study will be of interest to researchers in insect microbiology and symbiosis. However, several issues need to be addressed before the manuscript can be considered for publication.

Major concerns:

1. The authors mainly describe the dynamics of bacterial communities rather than exclusively endosymbiont communities. Therefore, I recommend changing the word Endosymbiont in the title to Bacterial.

2. OTUs typically represent sequence clusters with 97% similarity, whereas ASVs denote exact sequence variants with 100% similarity, offering finer taxonomic resolution. Relying on OTUs rather than ASVs may obscure subtle microbial differences and raises concerns about the statistical robustness and reproducibility of the analysis.

3. When comparing microbial composition among treatments, I recommend using PERMANOVA analysis and presenting the results in Figure 2. In addition, in Figure 6, the Chinese word “分类” should be replaced with the English term taxonomy.

4. Wolbachia is the dominant microbial taxon in the CK group but declines markedly in Groups A and B. The discussion should address whether this pattern has been observed in other species and explore possible biological mechanisms underlying this phenomenon.

Minor comments:

Line 12  Use “leaf-feeding” instead of “leaf-eating.”

Line 14  The term Endosymbiotic does not need to be italicized.

Lines 125-135  Please clarify whether the larvae were surface-sterilized before homogenization. Otherwise, surface microbes may have influenced the results.

Lines 142-152  Indicate whether the data were rarefied before diversity analyses.

Line 158  There is a redundant period; please correct the punctuation.

Line 160  The use of QIIME 1.9.1 is outdated. Re-analysis using more current platforms (e.g., QIIME 2 or DADA2) is recommended for improved taxonomic resolution and reproducibility.

Line 163  The letter “P” should be italicized.

Lines 167-176  The sequence accession numbers are not part of the main results and should be moved to the Data Availability Statement section.

Line 340  The word “and” does not need to be italicized.

Lines 348-349  This statement lacks supporting references; please provide appropriate citations (for example: https://doi.org/10.1111/1758-2229.70013).

Line 351  There appears to be a typographical error in the phrase “an imbal, an imbalancet”; please check and correct it.

In Figure 1, the first letter of the title of the vertical axis should be capitalized, and each sub-figure should be clearly labeled with letters a, b, c .

Author Response

Comments 1: The authors mainly describe the dynamics of bacterial communities rather than exclusively endosymbiont communities. Therefore, I recommend changing the word Endosymbiont in the title to Bacterial.

Response 1: Thank you for this insightful suggestion. We agree that the term "bacterial communities" more accurately reflects the broad scope of our study, which encompasses the successional dynamics of both endogenous and environmentally acquired taxa, rather than being strictly limited to obligate endosymbionts. We have revised the title (and any other relevant instances throughout the manuscript) accordingly by replacing "Endosymbiont" with "Bacterial" to ensure precision and clarity.

Comments 2: OTUs typically represent sequence clusters with 97% similarity, whereas ASVs denote exact sequence variants with 100% similarity, offering finer taxonomic resolution. Relying on OTUs rather than ASVs may obscure subtle microbial differences and raises concerns about the statistical robustness and reproducibility of the analysis. When comparing microbial composition among treatments, I recommend using PERMANOVA analysis and presenting the results in Figure 2. In addition, in Figure 6, the Chinese word “分类” should be replaced with the English term taxonomy.

Response 2: Indeed, there are several newer algorithms for microbial data processing (such as ASV, etc.). However, the advantages and disadvantages of OTU and ASV methods remain debated, and OTU clustering is still widely adopted in most publications. We also conducted analysis using ASV (implemented in QIIME 2) as an alternative to OTU. The results showed consistent major microbial trends (we have provided the results obtained with the new method in our response). Thus, no modifications have been made to the methodology or results section.  Meanwhile, following the reviewer's suggestion, we used permutational multivariate analysis of variance (PERMANOVA) to compare the microbial composition among different treatment groups and have presented the results in Figure 2. Additionally, we have revised Figure 6 accordingly.

Comments 3: Wolbachia is the dominant microbial taxon in the CK group but declines markedly in Groups A and B. The discussion should address whether this pattern has been observed in other species and explore possible biological mechanisms underlying this phenomenon.

Response 3: Thank you for your review comments. The observed pattern has not been widely reported in the literature. Although occasional studies have indicated a decline in Wolbachia abundance under specific experimental or environmental conditions, no consistent pattern has been identified across different host species. Therefore, this phenomenon may reflect a unique aspect of the system under investigation. The underlying biological mechanisms may involve complex host-symbiont interactions and evolutionary strategies, potentially including host immune responses and competition with other microbiota. However, due to the current lack of consistent evidence across species, it is not possible to identify a universal mechanism. We have added relevant discussion in the revised manuscript to elaborate on this perspective.

Comments 4: Minor comments:

Line 12  Use “leaf-feeding” instead of “leaf-eating.”

Line 14  The term Endosymbiotic does not need to be italicized.

Line 158  There is a redundant period; please correct the punctuation.

Line 163  The letter “P” should be italicized.

Lines 167-176  The sequence accession numbers are not part of the main results and should be moved to the Data Availability Statement section.

Line 340  The word “and” does not need to be italicized.

Lines 348-349  This statement lacks supporting references; please provide appropriate citations (for example: https://doi.org/10.1111/1758-2229.70013).

Line 351  There appears to be a typographical error in the phrase “an imbal, an imbalancet”; please check and correct it

Response 4: We thank the reviewer for their careful reading and helpful corrections. We have addressed all the typographical errors and formatting issues as suggested.

Comments 5: Lines 125-135  Please clarify whether the larvae were surface-sterilized before homogenization. Otherwise, surface microbes may have influenced the results.

Response 5: We thank the reviewer for this insightful comment. The reviewer raises a critical point. Indeed, the larvae were not surface-sterilized prior to homogenization. We fully acknowledge that this may have resulted in some surface microbes contributing to the observed microbial community. Our rationale was primarily based on two aspects: First, all experimental groups were reared and handled under identical conditions. Therefore, we believe the background of surface microbiota was relatively consistent and controllable across all samples. This consistency ensures the validity of inter-group comparisons, meaning that the observed differences are more likely to stem from the internal treatments rather than external contamination. Second, a key objective of this study is to simulate the natural decomposition process of insects after death, which inherently involves the invasion and participation of surface microbes. Thus, retaining the surface microbiota aligns more closely with our experimental goal of simulating a realistic scenario.

Comments 6: Lines 142-152  Indicate whether the data were rarefied before diversity analyses.

Response 6: Thanks for your constructive comments. The data used for our alpha diversity analysis were not rarefied, as rarefaction may lead to data loss, potentially introduce new biases, and reduce statistical power.

Comments 7: Line 160  The use of QIIME 1.9.1 is outdated. Re-analysis using more current platforms (e.g., QIIME 2 or DADA2) is recommended for improved taxonomic resolution and reproducibility.

Response 7: Thank you for your review comments. We have also performed an analysis using ASV (in place of OTU) through QIIME 2. The results demonstrated consistent major microbial trends (we have included the results obtained with this new method in our response). Therefore, no changes have been made to the Methods or Results sections.

Comments 8: In Figure 1, the first letter of the title of the vertical axis should be capitalized, and each sub-figure should be clearly labeled with letters a, b, c .

Response 8: Thanks for your valuable suggestion. We have addressed all the typographical errors and formatting issues as suggested.

Round 2

Reviewer 1 Report

Comments and Suggestions for Authors

Overall, the reviewers have addressed my concerns. I have a few minor corrections:

105: Italics for Bacteroidetes
149: Please provide detail about where the carcasses incubated -  outside of a tube? In the lab environment or natural environment?

"bacteria" and "bacterial" do not need to be capitalized

 Thank you for explaining why Day 7 and Day 21 time points were selected. It would help the reader if the explanation provided in the response letter were to be placed in the text.

Author Response

Comments 1: 105: Italics for Bacteroidetes

"bacteria" and "bacterial" do not need to be capitalized

Thank you for explaining why Day 7 and Day 21 time points were selected. It would help the reader if the explanation provided in the response letter were to be placed in the text.

 Response 1: Thank you very much for your suggestion. I have revised the manuscript according to your advice.

Comments 2: 149: Please provide detail about where the carcasses incubated -  outside of a tube? In the lab environment or natural environment?

Response 2: We thank the reviewer for this valuable suggestion. The larval were statically incubated in sealed tubes placed in a laboratory incubator at 23°C to simulate natural decomposition. We have now incorporated this clarification into the main text of the manuscript.